# BERENICE Final Analysis: Cardiac Safety Study of Neoadjuvant Pertuzumab, Trastuzumab, and Chemotherapy Followed by Adjuvant Pertuzumab and Trastuzumab in HER2-Positive Early Breast Cancer

**DOI:** 10.3390/cancers14112596

**Published:** 2022-05-24

**Authors:** Chau Dang, Michael S. Ewer, Suzette Delaloge, Jean-Marc Ferrero, Ramon Colomer, Luis de la Cruz-Merino, Theresa L. Werner, Katherine Dadswell, Mark Verrill, Daniel Eiger, Sriparna Sarkar, Sanne Lysbet de Haas, Eleonora Restuccia, Sandra M. Swain

**Affiliations:** 1Department of Medicine, Breast Medicine Service, Memorial Sloan Kettering Cancer Center, New York, NY 10065, USA; 2Department of Cardiology, The University of Texas MD Anderson Cancer Center, Houston, TX 77030, USA; mewer@mdanderson.org; 3Department of Medical Oncology, Institut Gustave Roussy, 94805 Paris, France; suzette.delaloge@gustaveroussy.fr; 4Department of Medical Oncology, Centre Antoine Lacassagne, University Côte d’Azur, 06110 Nice, France; jean-marc.ferrero@nice.unicancer.fr; 5Division of Medical Oncology, Hospital Universitario La Princesa, 28006 Madrid, Spain; rcolomer@seom.org; 6Department of Clinical Oncology, Hospital Universitario Virgen Macarena, 41009 Seville, Spain; ldelacruzmerino@gmail.com; 7Huntsman Cancer Institute, University of Utah, Salt Lake City, UT 84112, USA; theresa.werner@hci.utah.edu; 8Global Product Development, Roche Products Limited, Welwyn Garden City AL7 1TW, UK; katherine.dadswell@roche.com; 9Northern Centre for Cancer Care, Freeman Hospital, Newcastle-upon-Tyne NE7 7DN, UK; mark.verrill@nhs.net; 10Product Development Oncology, F. Hoffmann-La Roche Ltd., 4070 Basel, Switzerland; daniel.eiger@roche.com (D.E.); eleonora.restuccia@roche.com (E.R.); 11External Business Partner, Roche Products Limited, Welwyn Garden City AL7 1TW, UK; sriparna.sarkar@businesspartner.roche.com; 12Oncology Biomarker Development, F. Hoffmann-La Roche Ltd., 4070 Basel, Switzerland; sanne_lysbet.de_haas@roche.com; 13Georgetown University Medical Center, Lombardi Comprehensive Cancer Center and MedStar Health, Washington, DC 20007, USA; sms248@georgetown.edu

**Keywords:** pertuzumab, trastuzumab, neoadjuvant, cardiac safety, early breast cancer

## Abstract

**Simple Summary:**

A combination of pertuzumab, trastuzumab, and chemotherapy is a standard treatment for patients with a type of breast cancer called HER2-positive. Before the BERENICE study, little was known about the safety and effectiveness of pertuzumab with trastuzumab after surgery. Cardiac safety was a particular concern, especially when the chemotherapy given before surgery included drugs called anthracyclines. BERENICE was designed to assess the cardiac safety of pertuzumab with trastuzumab before surgery in combination with two different types of anthracycline-based chemotherapies. This paper describes additional safety and effectiveness data from BERENICE after patients had undergone surgery and when they had finished treatment. The incidence of cardiac side effects was low regardless of anthracycline use. The cardiac safety of pertuzumab and trastuzumab is now well characterized based on our study and others. Available information supports the use of pertuzumab–trastuzumab-based therapies as a standard treatment in HER2-positive early breast cancer.

**Abstract:**

BERENICE (NCT02132949) assessed the cardiac safety of the neoadjuvant–adjuvant pertuzumab–trastuzumab-based therapy for high-risk, HER2-positive early breast cancer (EBC). We describe key secondary objectives at final analysis. Eligible patients received dose-dense doxorubicin and cyclophosphamide q2w × 4 ➝ paclitaxel qw × 12 (Cohort A) or 5-fluorouracil, epirubicin, cyclophosphamide q3w × 4 ➝ docetaxel q3w × 4 (B) as per physician’s choice. Pertuzumab–trastuzumab (q3w) was initiated from the taxane start and continued post-surgery to complete 1 year. Median follow-up: 64.5 months. There were no new cardiac issues and a low incidence of Class III/IV heart failure (Cohort B only: one patient (0.5%) in the adjuvant and treatment-free follow-up (TFFU) periods). Fourteen patients (7.7%) had LVEF declines of ≥10% points from baseline to <50% in Cohort A, as did 20 (10.5%) in B during the adjuvant period (12 (6.2%) in A and 7 (3.6%) in B during TFFU). The five-year event-free survival rates in Cohorts A and B were 90.8% (95% CI: 86.5, 95.2) and 89.2% (84.8, 93.6), respectively. The five-year overall survival rates were 96.1% (95% CI: 93.3, 98.9) and 93.8% (90.3, 97.2), respectively. The final analysis of BERENICE further supports pertuzumab–trastuzumab-based therapies as standard of care for high-risk, HER2-positive EBC.

## 1. Introduction

Since the approval of pertuzumab together with trastuzumab and chemotherapy for the treatment of patients with human epidermal growth factor receptor 2 (HER2)-positive metastatic breast cancer [1], followed by its incorporation into the early breast cancer (EBC) setting [2], much has evolved and is still unfolding. Though survival outcomes of patients with HER2-positive breast cancer have greatly improved by adding pertuzumab to trastuzumab and chemotherapy, long-term toxicities, mainly cardiotoxicities, have emerged as a particular concern [3]. When the BERENICE study (NCT02132949) was designed, there were limited data for understanding the potential interaction(s) of anthracycline-based chemotherapy with pertuzumab and trastuzumab, particularly the cardiac safety. Long-term efficacy data from NeoSphere [4] and TRYPHAENA [5] were promising; however, both studies involved treatment with a single anti-HER2 agent, trastuzumab, in the adjuvant setting, following dual HER2 blockade with pertuzumab and trastuzumab in the neoadjuvant setting. Clinical trial data for the safety and efficacy of the continuation of dual HER2 blockade with pertuzumab and trastuzumab across the neoadjuvant and adjuvant settings were lacking at the time this regimen was approved in both settings.

To address these knowledge gaps, the phase II BERENICE trial was designed to incorporate pertuzumab in both the neoadjuvant and adjuvant settings, to complete 1 year of dual anti-HER2 therapy. The trial comprised two cohorts: one with dose-dense doxorubicin plus cyclophosphamide (ddAC) followed by paclitaxel in combination with trastuzumab and pertuzumab (Cohort A), and another with a conventional schedule anthracycline-based chemotherapy: fluorouracil, epirubicin, and cyclophosphamide (FEC) (Cohort B) followed by docetaxel with trastuzumab and pertuzumab in the neoadjuvant setting at the investigator’s discretion [6]. At the primary analysis, BERENICE met its primary endpoint by showing an overall low incidence of New York Heart Association (NYHA) Class III/IV heart failure in both cohorts during the neoadjuvant period: 1.5% in Cohort A and none in Cohort B [6]. Cardiac safety was assessed by (1) the incidence of NYHA Class III/IV heart failure and (2) left ventricular ejection fraction (LVEF) declines of ≥10 percentage points from baseline and to a value of <50%, measured by an echocardiogram or a multigated acquisition scan. Few patients experienced ≥1 LVEF decline of ≥10 percentage-points from baseline and to a value of <50% (6.5% in Cohort A and 2.0% in Cohort B). General safety and cardiac safety were consistent with the known pertuzumab plus trastuzumab and chemotherapy profiles [7,8]. The total pathologic complete response (tpCR; ypT0/is ypN0) rates in BERENICE were 61.8% and 60.7% in Cohorts A and B, respectively [6], in line with the clinical activity of pertuzumab-based neoadjuvant therapies previously demonstrated [7,8].

In this final analysis of BERENICE, we report safety data from the adjuvant and treatment-free follow-up (TFFU) periods, as well as secondary long-term efficacy endpoints: event-free survival (EFS), invasive disease-free survival (IDFS), and overall survival (OS).

## 2. Materials and Methods

### 2.1. Study Design and Patients

Details of the BERENICE study have been published previously [6]. Briefly, BERENICE was a phase II, open-label, multicenter, multinational, noncomparative, two-cohort cardiac safety study conducted across 75 centers in 12 countries. Key eligibility criteria included centrally confirmed HER2-positive, locally advanced, inflammatory, or early stage, unilateral, and histologically confirmed invasive breast cancer, no prior breast or systemic cancer history (within 5 years), no uncontrolled systemic disease, good cardiac status (patients were excluded if they had poorly controlled hypertension (e.g., systolic blood pressure > 180 mmHg and/or diastolic blood pressure > 100 mmHg), angina requiring antianginal medication, history of congestive heart failure of any NYHA classification, serious or uncontrolled cardiac arrhythmia requiring treatment, or history of myocardial infarction within 6 months of enrollment), a baseline LVEF ≥ 55%, an Eastern Cooperative Oncology Group performance status ≤ 1, and no prior incisional biopsy/excision of the primary tumor. BERENICE was conducted in full accordance with the guidelines for Good Clinical Practice and the Declaration of Helsinki. Written informed consent was obtained from each participant. Approval for the protocol and for any modifications was obtained from independent ethics committees.

### 2.2. Procedures

In the neoadjuvant period, patients in Cohort A received four cycles of ddAC every two weeks (q2w); 60 mg/m^2^ doxorubicin and 600 mg/m^2^ cyclophosphamide with granulocyte-colony stimulating factor support as needed, followed by 12 qw doses of paclitaxel (80 mg/m^2^) in combination with four three-weekly (q3w) cycles of trastuzumab and pertuzumab (trastuzumab at an 8 mg/kg loading dose followed by 6 mg/kg maintenance doses; pertuzumab at an 840 mg loading and 420 mg maintenance doses). In Cohort B, patients received four q3w cycles of FEC (500 mg/m^2^/100 mg/m^2^/600 mg/m^2^) followed by four q3w cycles of docetaxel (75 mg/m^2^, escalated to 100 mg/m^2^, if tolerated) plus four q3w cycles of trastuzumab (8 mg/kg loading dose; 6 mg/kg maintenance dose, q3w) and pertuzumab (840 mg/kg loading dose; 420 mg/kg maintenance dose, q3w). All were given intravenously.

Patients in both cohorts were scheduled to undergo surgery after eight cycles of neoadjuvant chemotherapy (after approximately 20 weeks in Cohort A, and approximately 24 weeks in Cohort B). Patients underwent breast-conserving surgery or mastectomy according to routine clinical practice. After a recovery period of at least 2 weeks, adjuvants pertuzumab and trastuzumab were given for a further 13 cycles intravenously (q3w) to complete a total of 17 cycles of anti-HER2 therapy. Treatment was continued until progression or recurrence of disease or unmanageable toxicity. Adjuvant hormone therapy and/or radiotherapy were given as clinically indicated according to local guidelines. Antibody-dose modifications were not permitted but treatments could be delayed if indicated. Patients could withdraw consent at any time, or could be withdrawn by the investigator or sponsor for safety reasons, if withdrawal was in the patient’s best interests, or for noncompliance. During the adjuvant period, adverse events (AEs; graded according to National Cancer Institute—Common Terminology Criteria for Adverse Events (NCI-NCTAE) Version 4.0), serious AEs (SAEs), lab abnormalities (graded according to NCI-NCTAE Version 4.0), and serum levels of antitherapeutic antibodies against pertuzumab were reported. Only drug-related SAEs, heart failure, pregnancies, and nonbreast-related second primary malignancies (irrespective of causal relationship) were reported during the TFFU period (defined as >42 days after the last dose of study drug, or one day after surgery, whichever was later, until the end of the study). All patients were followed up for a period of approximately 5 years after enrollment of the last patient, even if their assigned treatment was discontinued early.

### 2.3. Assessments

Tumors were assessed at screening/baseline and at each neoadjuvant cycle. Patients were assessed for recurrence at cycle 9, cycle 13, cycle 17, cycle 21, and at treatment completion. Chemotherapy was given for the first 4 cycles; HER2 therapy began at cycle 5 and was given for a total of 17 cycles. LVEF assessments were conducted at screening/baseline, and within 3 days prior to day 1 of each of the following treatment cycles: 5, 7 (neoadjuvant period), 9, 12, 15, 18 (adjuvant period), at treatment completion and every 6 months for 2 years, then annually for a further 2 years after the completion of therapy. Confirmed LVEF assessments were defined as LVEF declines of ≥10% points from baseline to <50% at two consecutive cardiac evaluations within 3 weeks by echocardiography or multiple-gated acquisition.

AEs and SAEs were assessed continuously using NCI-NCTAE Version 4.0. The causality of AEs with study treatment was assessed by the investigator. The study was open-label; safety data were reviewed at regular intervals by the study steering committee, which included representatives of the sponsor and study investigators as well as an independent cardiology expert (MSE).

Gene expression (RNA) was assessed by a custom 800-gene codeset on the Nanostring nCounter platform using baseline biopsy samples provided for central HER2 testing during the study screening period. The panel of genes required to assess intrinsic breast cancer subtypes, the prediction analysis of microarray 50 (PAM50) [9] was included within the set of genes analyzed. The PAM50 subtype prediction to describe major intrinsic subtypes (luminal A, luminal B, HER2-enriched, and basal-like) was carried out using a random-forest-based classifier [10].

### 2.4. Statistical Analysis

The results are descriptive; no statistical hypothesis testing was planned. The primary objective (cardiac safety during the neoadjuvant period) has been described previously [6]. Sample sizes per cohort were calculated to establish, with an acceptable precision on the basis of exact Clopper–Pearson 95% confidence intervals (CIs), the expected rates of NYHA Class III/IV heart failure and LVEF decline. The exact CIs of expected rates (NYHA Class III/IV heart failure rate < 3% and LVEF decline rate of ≤6%) [7,8] were used to evaluate cardiac safety. Safety analyses were performed on all patients who received at least one dose of study medication. Secondary safety objectives included cardiac safety during the adjuvant period and TFFU period, and general safety during the study. The main secondary efficacy objective (tpCR rate) has been described previously [6]. Other secondary efficacy endpoints included EFS, IDFS, and OS. EFS was defined as the time from enrollment to the first occurrence of disease progression or relapse (excluding ipsilateral or contralateral in situ disease and second primary nonbreast cancers) or death from any cause. IDFS was defined as the time from the first date of no disease (i.e., the date of primary surgery) to the first occurrence of progressive invasive disease or relapse (excluding ipsilateral or contralateral in situ disease and second primary nonbreast cancers), or death from any cause. This definition of IDFS (which excludes second primary nonbreast cancers as events) differs from the standardized definitions for efficacy endpoints definition [11] and was preplanned in the protocol. Note that the neoadjuvant treatment period was 4 weeks longer in Cohort B compared with Cohort A in this nonrandomized study, hence the data between cohorts should not be compared directly. OS was defined as the time from enrollment to death from any cause. The Kaplan–Meier approach was used to plot EFS, IDFS, and OS, as well as to estimate the proportion of patients who were event-free at landmark timepoints for each cohort. Efficacy was analyzed in the intention-to-treat (ITT) population (all enrolled patients) and is descriptive only. The assessment of the tpCR rate according to PAM50 subtype [6] was a predefined exploratory objective. Ad hoc exploratory analyses included EFS by central hormone receptor status, nodal status, neoadjuvant response (tpCR status), and PAM50 subtype.

## 3. Results

### 3.1. Patients

Patients were enrolled between July 2014 and August 2015. Clinical cutoff for the adjuvant analysis was 7 January 2017 (once all patients had completed adjuvant therapy) and for the final analysis, 25 August 2020 (last patient visit during TFFU). The ITT populations were 199 and 201 patients for Cohort A and Cohort B, respectively. The safety populations were 199 and 198 patients, respectively, and 182 and 189 patients completed neoadjuvant treatment [6]. A total of 163 and 176 patients completed 17 cycles of anti-HER2 therapy, respectively; 195 patients in each cohort started TFFU. Overall, 114 and 147 patients completed the study (Appendix A). The most common reason for study discontinuation was nonsafety (76/199 patients (38.2%) in Cohort A and 37/198 patients (18.7%) in Cohort B), of which most were due to noncompletion of the protocol-defined study follow-up period (5 years after the last patient enrolled; 44/199 patients (22.1%) in Cohort A and 26/198 patients (13.1%) in Cohort B). The patient demographics and baseline characteristics for the ITT population have been described previously and were generally balanced between cohorts. However, Cohort A had more patients who were classified as obese (body mass index (BMI) ≥ 30) than Cohort B (27.4% and 17.3%), and Cohort A had more patients who had concurrent hypertension than Cohort B (27.6% and 15.2%). The median age was 49.0 years in each cohort [6]. The median follow-up was 64.1 months (95% CI: 63.0, 64.9) for Cohort A and 64.8 months (95% CI: 64.4, 65.2) for Cohort B at the end of the study.

Baseline characteristics in the PAM50-evaluable population were generally well balanced, although there were slightly more patients with HER2-enriched disease in Cohort B (data not shown).

### 3.2. Cardiac Safety

The cardiac safety during the neoadjuvant period has been published previously [6]. Table 1 summarizes the incidence of cardiac events by trial period (neoadjuvant, adjuvant, and TFFU period).

#### 3.2.1. Adjuvant Period

During the adjuvant period, no patients in Cohort A experienced an NYHA Class III/IV heart failure event; one patient in Cohort B (0.5%) experienced one event (Table 1). Fourteen patients in Cohort A (7.7%) and 20 patients in Cohort B (10.5%) had ≥1 LVEF decline of ≥10% points from baseline to <50%. A confirmed LVEF decline (at least two consecutive LVEF declines of ≥10% points from baseline to <50%) was reported in five patients (2.8%) in Cohort A and six patients (3.2%) in Cohort B.

#### 3.2.2. TFFU Period

During the TFFU period, no patients in Cohort A experienced an NYHA Class III/IV heart failure event; one patient in Cohort B (0.5%) experienced one event (Table 1). The assessment of LVEF decrease during the TFFU period showed that 12 patients in Cohort A (6.0%) and 7 patients in Cohort B (3.5%) had ≥1 LVEF decline of ≥10% points from baseline to <50% (Table 1). A confirmed LVEF decline was reported in six patients (3.0%) in Cohort A versus two patients (1.0%) in Cohort B.

#### 3.2.3. Timing of Occurrence of Cardiac Events

In Cohort A, all three cases of NYHA Class III/IV heart failure occurred first during the neoadjuvant period, with no events observed in the adjuvant or TFFU periods, while in Cohort B one case occurred during the adjuvant period and one during the TFFU period (Table 1). In Cohort A, a similar number of LVEF declines of ≥10% points from baseline to <50% were observed during the neoadjuvant, adjuvant, and TFFU periods; while in Cohort B, most cases occurred during the adjuvant period followed by the TFFU period and the neoadjuvant period. Appendix A summarize the timing of occurrence of the first NYHA Class III/IV heart failure and timing of occurrence of the first LVEF decline of ≥10% points from baseline to <50%, respectively.

#### 3.2.4. Mean Change in LVEF from Baseline

The mean LVEF at baseline in both cohorts was ≥64.0% (95% CI: 54.0, 83.6). As expected, the mean LVEF dropped below baseline in both cohorts, with a maximum mean LVEF decrease of −6.1%. At the latest LVEF assessment (follow-up month 48 during the TFFU period), the mean LVEF had almost returned to baseline (decrease from baseline of −2.3% and −2.0% in Cohorts A and B, respectively) (Figure 1).

#### 3.2.5. Cardiac Event Resolution Rates

Of all the LVEF declines observed (either by <10% points or by ≥10% points from baseline to <50%) 92.3% (24/26) had resolved in Cohort A and 100% (26/26) had resolved in Cohort B. The resolution rates for NYHA Class II–IV heart failure events were 80.0% (4/5) in Cohort A and 100% (3/3) in Cohort B.

### 3.3. General Safety

The safety during the neoadjuvant period has been published previously [6].

Most patients (83.3% in Cohort A, and 80.0% in Cohort B) completed all 13 cycles of pertuzumab and trastuzumab during adjuvant treatment; patients were treated for a median of 39 weeks after surgery in both cohorts (Appendix A). Most pertuzumab infusions were given without dose delays or interruptions. Similar numbers of patients in both cohorts had pertuzumab delayed, interrupted, or discontinued (Cohort A, 27.2%; Cohort B, 26.3%). Dose delays/interruptions due to an AE were observed in 8.9% and 13.2% of patients in Cohort A and Cohort B, respectively (Appendix A). Of those patients who did experience a dose delay/interruption, the majority did so for only one cycle of treatment. As expected, trastuzumab exposure data paralleled that of pertuzumab (Appendix A).

#### 3.3.1. Adjuvant Period

Most patients experienced at least one AE (≥90.0% patients in both cohorts) (Table 2).

A total of 5.0% of patients in Cohort A and 5.8% in Cohort B withdrew from pertuzumab or trastuzumab treatment due to an AE. The most common all-grade AEs (≥20% in either treatment arm) were arthralgia, diarrhea, and radiation skin injury. Notable differences between cohorts were all-grade diarrhea and all-grade mucositis. Most AEs were grades 1 or 2 in both cohorts (81.8% in Cohort A and 69.0% in Cohort B). The proportion of patients experiencing a grade ≥3 AE was 12.7% in Cohort A and 21.1% in Cohort B. The most frequently reported grade ≥3 AEs (≥2% in either cohort) were ejection fraction decreased (2.8% of patients in Cohort A and 3.2% of patients in Cohort B), gastroenteritis (0% and 2.1%, respectively), radiation skin injury (0.6% and 2.1%, respectively), and hypertension (0.6% and 2.1%, respectively). SAEs were reported in 8.3% of patients in Cohort A and 8.9% in Cohort B. SAEs reported by ≥2 patients in any cohort included ejection fraction decreased (1.1% in Cohort A and 2.1% in Cohort B) and mastitis (1.1% in Cohort A and 0% in Cohort B). A total of 58.0% of patients in Cohort A and 64.2% of patients in Cohort B experienced ≥1 AE that was suspected to be causally related to the pertuzumab or trastuzumab treatment, the most common of which were diarrhea (7.2% in Cohort A and 18.4% in Cohort B) and ejection fraction decreased (7.7% in Cohort A and 10.0% in Cohort B). No deaths occurred during the adjuvant period.

#### 3.3.2. TFFU Period

During the TFFU period, the incidence of drug-related grade ≥ 3 AEs (1.0% and 2.5%) and SAEs (1.5% and 3.5%) were low in Cohorts A and B, respectively. No patients in Cohort A and six patients (3.0%) in Cohort B experienced a second nonbreast primary malignancy. These were two plasma cell myelomas (1.0%), one adenocarcinoma of the colon (0.5%), one basal cell carcinoma (0.5%), one colon cancer (0.5%), and one non-Hodgkin’s lymphoma (0.5%). There were 7 deaths in Cohort A (3.5%) and 13 in Cohort B (6.6%). The primary cause of death was disease progression (4 (2.0%) and 12 (6.1%) in Cohorts A and B, respectively). All deaths in the study occurred during the TFFU period (Table 3).

### 3.4. Efficacy

Fifteen patients (7.5%) in Cohort A and 25 (12.6%) in Cohort B experienced disease recurrence. Distant recurrences were the most frequent events (6.5% and 6.6% in Cohorts A and B, respectively); within this category, central nervous system (CNS) recurrence was low in both cohorts (2.0% and 3.5%, respectively) (Table 4). Local recurrences were seen in 1.0% and 2.0% of patients, respectively; the same proportions were observed for regional recurrences. A second primary invasive breast cancer was observed in 1.0% of patients in each cohort. A second primary nonbreast malignancy was observed in 0.5% of patients in Cohort A and 3.0% in Cohort B.

Nanostring data were available for 339 tumors; 294 of these could be classified with a PAM50 subtype (148 in Cohort A and 146 in Cohort B) [6], of which 59.5% were HER2-enriched. The highest tpCR rates were in HER2-enriched tumors (75.0% and 73.7% for Cohort A and Cohort B, respectively) [6].

#### 3.4.1. Five-Year EFS

The 5-year EFS rate in Cohort A was 90.8% (95% CI: 86.5, 95.2) and 89.2% (95% CI: 84.8, 93.6) in Cohort B (Figure 2a). The EFS for key subgroups and by PAM50 status are shown in Figure 2b–d and Appendix A, respectively; these data should be interpreted with caution due to the low patient numbers. In Cohort A, the EFS rates for patients with hormone-receptor-positive disease (*n* = 129) and hormone-receptor-negative disease (*n* = 65) were 89.9% (95% CI: 84.1, 95.7) and 92.0% (95% CI: 85.3, 98.7), respectively. In Cohort B, the EFS rates were 94.2% (*n* = 124; 95% CI: 90.0, 98.4) and 82.3% (*n* = 75; 95% CI: 73.5, 91.0), respectively (Figure 2b). In Cohort A, the EFS rates for patients who achieved a tpCR following neoadjuvant therapy compared with those who had residual disease were 93.2% (*n* = 123; 95% CI: 88.6, 97.8) and 86.7% (*n* = 76; 95% CI: 77.7, 95.6), respectively. In Cohort B, the EFS rates were 94.2% (*n* = 122; 95% CI: 90.0, 98.4) and 81.2% (*n* = 79; 95% CI: 72.3, 90.1), respectively (Figure 2d).

#### 3.4.2. Four-Year IDFS

The 4-year IDFS rate was 92.6% (95% CI: 88.7, 96.5) in Cohort A and 91.1% (95% CI: 87.0, 95.1) in Cohort B (Figure 2e).

#### 3.4.3. Five-Year OS

The OS rate in Cohort A was 96.1% (95% CI: 93.3, 98.9) and 93.8% (95% CI: 90.3, 97.2) in Cohort B (Figure 2f).

## 4. Discussion

Since the advent of HER2-targeted therapy, cardiotoxicity has been the class-defining safety concern. Not only is HER2 signaling essential for maintaining the homeostasis of the HER2-rich cardiomyocytes [12], it is also required for surviving the oxidative stress elicited by anthracycline chemotherapies [13], which ultimately leads to dilated cardiomyopathy when trastuzumab is given concomitantly with doxorubicin [14]. With an excess of 5 years of follow-up and regular LVEF assessments, BERENICE provides reassuring long-term cardiac safety data for sequential therapy with anthracycline-based chemotherapies followed by anti-HER2 therapy. It also demonstrates the feasibility of the dose-dense regimen, by showing comparably low rates of NYHA Class III/IV heart failure (1.5% in Cohort A and 1.0% in Cohort B) and LVEF declines of ≥10% from baseline to <50% (13.6% in Cohort A and 12.1% in Cohort B) overall. These data should be interpreted with caution for this nonrandomized safety study.

Though it was hypothesized that dual HER2 blockade, by eliciting additional inhibition of HER2 signaling, could increase the cardiotoxicity of HER2-positive BC treatment compared with trastuzumab alone, this has never been demonstrated in diverse clinical trials and meta-analyses [4,15,16,17]. The BERENICE study further adds to the body of evidence that dual HER2-blockade with pertuzumab plus trastuzumab is feasible from a cardiotoxicity standpoint. The continuation of dual HER2 blockade with pertuzumab and trastuzumab from the neoadjuvant to the adjuvant setting has also been assessed in the KRISTINE trial (pertuzumab + trastuzumab versus pertuzumab + ado-trastuzumab emtansine) [18], and in the ongoing PEONY trial (pertuzumab + trastuzumab versus placebo + trastuzumab) [19], where a low incidence of cardiac events have been reported during the neoadjuvant treatment period; adjuvant treatment and long-term follow-up is ongoing. Data from the phase III FeDeriCa study assessing intravenous pertuzumab plus trastuzumab versus the fixed-dose combination of pertuzumab and trastuzumab for subcutaneous injection across the neoadjuvant and adjuvant settings also showed low rates of cardiac events in both arms [20].

APHINITY was the first study to report data for the use of 18 cycles of pertuzumab in combination with trastuzumab and chemotherapy in the adjuvant setting; incidence of primary cardiac events was also low (<1%) after a 45.4-month median follow-up [21], with only one further primary cardiac event (heart failure) observed in the pertuzumab arm since the primary analysis [15]. Cardiac safety was maintained after a 74-month median follow-up, with no further primary or secondary cardiac events in either arm [22]. In contrast to APHINITY, patients in BERENICE received all of their chemotherapy plus pertuzumab and trastuzumab before surgery, then continued pertuzumab plus trastuzumab after surgery.

Consistent with TRYPHAENA [5] and other trials investigating dual HER2 blockade with pertuzumab and trastuzumab [4,22], BERENICE has shown that once dual HER2 blockade treatment is completed, the incidence of clinically relevant cardiac events decreases, and there is a trend for LVEF recovery towards baseline. In this regard, the mechanistic on-target cardiotoxic effect of pertuzumab plus trastuzumab is evident and, most importantly, is generally reversible.

LVEF periodicity assessments performed currently in clinical trials and clinical practice hold the potential for false-positive low LVEF readings. Ejection fraction declines are not specific to the investigational therapies, but may be the result of metabolic changes, circulatory changes involving preload and afterload, or other ingested or natural agents. Additionally, the interpretation of cardiac ultrasound studies is imperfect due to inter- and intra-operator variation. In an effort to quantify these variables, the likelihood of a false-positive reading if four LVEF assessments are undertaken is approximately 3.6% [23]. This number will be higher if assessments are undertaken during long-term surveillance, or if the lower limit of normal for cardiac ultrasound is increased [23]. However, using confirmed LVEF decline data (at least two consecutive significant LVEF of <50% and a ≥10% decline from baseline) reduces the incidence of false-positive results.

No new general safety concerns arose during the long-term follow-up of the BERENICE study, with low incidences of drug-related grade ≥ 3 adverse events and serious adverse events in both cohorts. All-grade diarrhea and all-grade mucositis were more common with the docetaxel-containing regimen than with the paclitaxel-containing regimen. Most deaths were, as anticipated, due to disease progression.

The 5-year EFS rates in BERENICE were similar in Cohorts A and B (90.8% and 89.2%, respectively), as were the 4-year IDFS and 5-year OS rates. Of note is the very low incidence of CNS metastases observed with the pertuzumab plus trastuzumab regimen, comparable also with that seen in studies of adjuvant trastuzumab alone [24] or with chemotherapy [25]. The BERENICE 5-year data showed higher EFS rates (90.8% in Cohort A and 89.2% Cohort B, respectively) compared with the 5-year EFS data described in NeoSphere (86%; neoadjuvant pertuzumab plus trastuzumab and docetaxel followed by adjuvant trastuzumab) [4], although cross-trial comparisons should be interpreted with caution, given the differences in patient populations and study designs. Nonetheless, these results underscore the enhanced efficacy of a full HER2 axis blockade versus a single inhibition of HER2 in treating patients with HER2-positive BC. The 4-year IDFS results in BERENICE were in keeping with the 4-year IDFS data reported in APHINITY in a similar high-risk population [21], suggesting that 1 year of pertuzumab plus trastuzumab as part of a complete EBC treatment regimen is optimal for patients at high risk of recurrence, with consistent results irrespective of time of surgery.

The BERENICE study has some limitations, including some imbalances in baseline characteristics and the number of patients completing the TFFU period between the two cohorts. Cohort A had more patients who were classified as obese (BMI ≥ 30), and more patients who had concurrent hypertension than Cohort B. Other limitations of BERENICE are due to the study design. As it is nonrandomized, the efficacy data should be interpreted with caution and the contribution of each chemotherapy backbone to both the toxicity and efficacy observed could not be assessed. Dose-intense schedules, which include shortening intervals (dose-density) between treatment cycles, or by giving individual drugs sequentially rather than concurrently, have been shown to moderately reduce the 10-year risk of recurrence and death from breast cancer [26]. The additional efficacy impact of the dose-dense schedule in Cohort A could not be determined in BERENICE due to the study design.

Although not statistically powered to assess long-term outcomes, the BERENICE data contribute to the important knowledge base regarding the clinical profile of pertuzumab plus trastuzumab when given together during both the neoadjuvant and adjuvant periods for a total of 1 year of HER2-targeted therapy. Overall, the 5-year efficacy data are promising; however, the toxicity of chemotherapy, particularly cardiotoxicity, is still a concern during concomitant administration of the HER2-targeted therapies and chemotherapy. Therefore, the de-escalation of chemotherapy, for example by removing anthracyclines, is one of the next steps in advancing treatment for patients with HER2-positive EBC. This was first evaluated in the BCIRG-006 trial, which showed similar efficacy but with fewer acute toxicities with a non-anthracycline- vs. anthracycline-containing trastuzumab-based regimen [27]. Subsequently, anthracycline-free regimens have been adopted in a number of clinical trials, e.g., TRAIN-2 [28,29], PHERGain [30], DecreSCendo (NCT04675827), and COMPASS (NCT04266249), and are preferred over anthracycline-containing regimens in clinical breast cancer guidelines [31].

Another limitation was that cardiac safety in the BERENICE study was assessed through monitoring the incidences of heart failure and LVEF decline alone. Although LVEF assessments are widely established as a prognostic tool for heart failure in clinical studies, there are limitations with this method [32,33]. The incorporation of newer techniques, such as measuring global longitudinal strain, may help to overcome these limitations and provide more comprehensive cardiac monitoring [32,33].

## 5. Conclusions

The cardiac safety of the combination of pertuzumab and trastuzumab is now well characterized based on the long-term follow-up of BERENICE and other studies. Data support the use of pertuzumab–trastuzumab-based therapies as standard of care in HER2-positive EBC, whether in combination with standard or dose-dense anthracycline-based chemotherapy. The BERENICE data show a high proportion of patients (89.2–90.8%) with no disease recurrence at >5 years of follow-up and no safety concerns after 1 year of pertuzumab plus trastuzumab as part of their complete EBC treatment. This is particularly important in the HER2-positive EBC setting as we can now speak about true long-term survivors of the disease.

The low cardiac toxicity, favorable efficacy outcomes, and consistency of BERENICE with previous studies support the use of pertuzumab plus trastuzumab throughout the whole course of treatment across both neoadjuvant and adjuvant treatment periods as the standard of care for high-risk HER2-positive EBC.

## Figures and Tables

**Figure 1 cancers-14-02596-f001:**
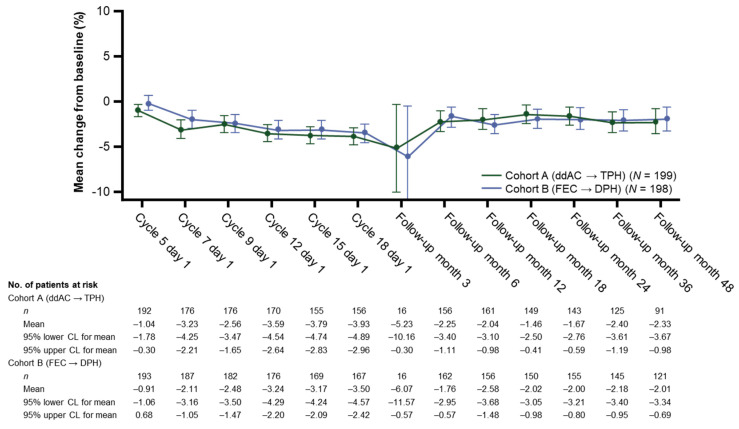
Mean change in left ventricular ejection fraction by visit. CL, confidence limit; ddAC, dose-dense doxorubicin plus cyclophosphamide; DPH, docetaxel, pertuzumab, and trastuzumab; FEC, fluorouracil, epirubicin, and cyclophosphamide; TPH, paclitaxel, pertuzumab, and trastuzumab.

**Figure 2 cancers-14-02596-f002:**
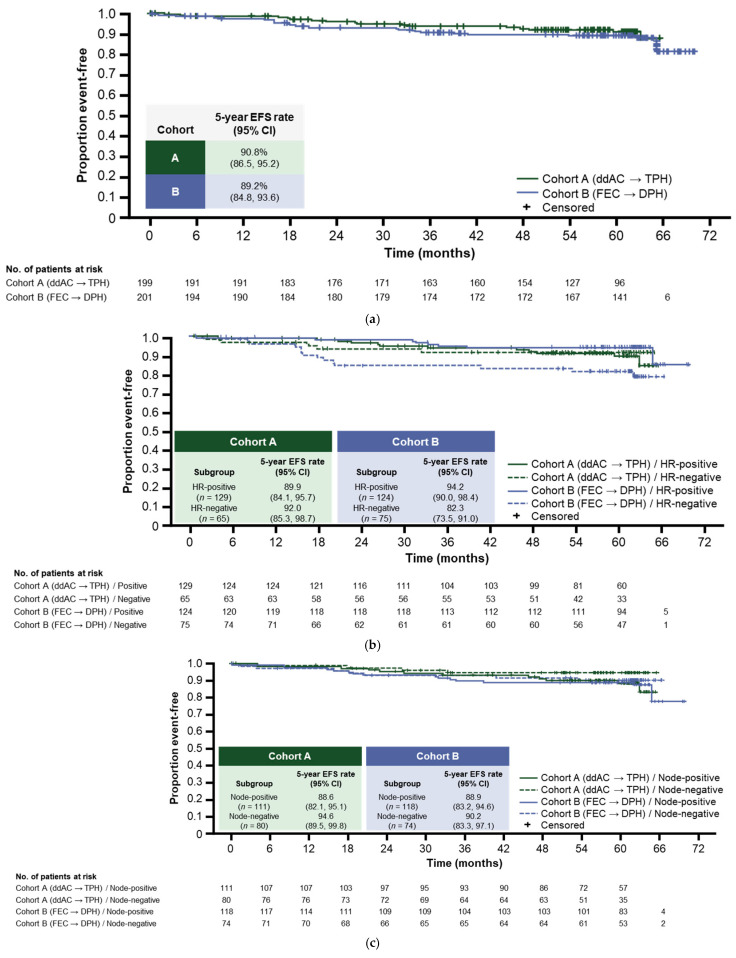
(**a**) EFS in the ITT population; (**b**) EFS in hormone receptor subgroups; (**c**) EFS by clinical nodal status; (**d**) EFS by tpCR status; (**e**) IDFS in the ITT population; (**f**) OS in the ITT population. CI, confidence interval; ddAC, dose-dense doxorubicin plus cyclophosphamide; EFS, event-free survival; DPH, docetaxel, pertuzumab, and trastuzumab; FEC, fluorouracil, epirubicin, and cyclophosphamide; HR, hormone receptor; IDFS, invasive disease-free survival; ITT, intention to treat; OS, overall survival; tpCR, total pathologic complete response (ypT0/is ypN0); TPH, paclitaxel, pertuzumab, and trastuzumab.

**Table 1 cancers-14-02596-t001:** Cardiac safety across all study periods.

Cardiac Safety Event	Cohort A (ddAC ➝ TPH)	Cohort B (FEC ➝ DPH)
**Class III/IV cardiac failure *n*, (%)**	3 (1.5)	2 (1.0)
Neoadjuvant period ^1^	3 (1.5)	0
Adjuvant period ^2^	0	1 (0.5)
TFFU period ^3^	0	1 (0.5)
**LVEF declines of ≥10% points from baseline to <50% *n*, (%); ^4^** **(confirmed events *n*, (%)) ^5^**	27 (13.6)	24 (12.1)
Neoadjuvant period	13 (6.5); (2 (1.0))	4 (2.0); (1 (0.5))
Adjuvant period	14 (7.7); (5 (2.8))	20 (10.5); (6 (3.2))
TFFU period	12 (6.0); (6 (3.0))	7 (3.5); (2 (1.0))

^1^ Neoadjuvant period safety population: Cohort A, *n* = 199; Cohort B, *n* = 198. ^2^ Adjuvant period safety population: Cohort A, *n* = 181; Cohort B, *n* = 190. ^3^ TFFU period safety population: Cohort A, *n* = 199; Cohort B, *n* = 198. ^4^ Includes symptomatic and asymptomatic events. Some patients experienced more than one LVEF decline of ≥10% points from baseline to <50% across different trial periods. ^5^ Defined as LVEF declines of ≥10% points from baseline to <50% at two consecutive cardiac evaluations by ECHO or MUGA. ddAC, dose-dense doxorubicin plus cyclophosphamide; DPH, docetaxel, pertuzumab, and trastuzumab; ECHO, echocardiography; FEC, fluorouracil, epirubicin, and cyclophosphamide; LVEF, left ventricular ejection fraction; MUGA, multiple-gated acquisition; TFFU, treatment-free follow-up; TPH, paclitaxel, pertuzumab, and trastuzumab.

**Table 2 cancers-14-02596-t002:** AEs during the adjuvant period.

	Adjuvant Phase	
**Total number of patients, *n* (%)**	**Cohort A (ddAC ➝ TPH)** ***n* = 181**	**Cohort B (FEC ➝ DPH)** ***n* = 190**
**≥1 AE**	171 (94.5)	171 (90.0)
Total AEs, *n*	1165	1237
**Deaths ^1^**	0	0
**Withdrawn due to an AE**		
Pertuzumab or trastuzumab	9 (5.0)	11 (5.8)
Pertuzumab	8 (4.4)	10 (5.3)
Any study drug	9 (5.0)	11 (5.8)
**Dose interruption/delay due to an AE**		
Pertuzumab or trastuzumab	23 (12.7)	30 (15.8)
Pertuzumab	22 (12.2)	28 (14.7)
**Grade ≥ 3 AE**	23 (12.7)	40 (21.1)
**Serious AE**	15 (8.3)	17 (8.9)
**AE suspected to be caused by pertuzumab or trastuzumab**	105 (58.0)	122 (64.2)
**AE suspected to be caused by pertuzumab**	104 (57.5)	120 (63.2)
**AE during pertuzumab or trastuzumab infusion**	3 (1.7)	1 (0.5)
**AE during pertuzumab infusion**	2 (1.1)	1 (0.5)
**AEs to monitor**		
Heart failure	0	1 (0.5)
Grade ≥ 3	0	0
Ejection fraction decreased	15 (8.3)	20 (10.5)
Grade ≥ 3	5 (2.8)	6 (3.2)
Diarrhea	26 (14.4)	45 (23.7)
Grade ≥ 3	0	2 (1.1)
Rash	59 (32.6)	61 (32.1)
Grade ≥ 3	0	2 (1.1)
Hypersensitivity/anaphylaxis	2 (1.1)	1 (0.5)
Grade ≥ 3	0	0
Mucositis	10 (5.5)	23 (12.1)
Grade ≥ 3	1 (0.6)	4 (2.1)
Leukopenia	12 (6.6)	12 (6.3)
Grade ≥ 3	0	1 (0.5)
Leukopenic infection	0	1 (0.5)
Grade ≥ 3	0	0
Febrile neutropenia	0	0
Grade ≥ 3	0	0
Febrile neutropenic infection	0	0
Grade ≥ 3	0	0
Interstitial lung disease	1 (0.6)	0
Grade ≥ 3	0	0
Infusion-related reactions ^2^	7 (3.9)	14 (7.4)
Grade ≥ 3	0	1 (0.5)

Percentages are of the total number of the safety-evaluated population entering the adjuvant treatment period as given in the column headings. Multiple occurrences of the same AE in one individual are counted only once except for “Total AEs”, in which multiple occurrences of the same AE are counted separately. The table includes AEs with onset from first dose of any study drug after surgery through to 42 days after the last dose of study drug. ^1^ Total number of deaths are counted over the adjuvant study treatment period. ^2^ As assessed by the investigator. AE, adverse events; ddAC, dose-dense doxorubicin plus cyclophosphamide; DPH, docetaxel, pertuzumab, and trastuzumab; FEC, fluorouracil, epirubicin, and cyclophosphamide; TPH, paclitaxel, pertuzumab, and trastuzumab.

**Table 3 cancers-14-02596-t003:** Summary of deaths.

Patients, *n* (%)	Cohort A (ddAC ➝ TPH) *n* = 199	Cohort B (FEC ➝ DPH) *n* = 198
**Deaths**	7 (3.5)	13 (6.6)
Disease recurrence	0	1 (0.5)
Disease progression	4 (2.0)	12 (6.1)
Adverse event	1 (0.5)	0
Other ^1^	2 (1.0)	0

^1^ One patient in Cohort A died due to COVID-19. ddAC, dose-dense doxorubicin plus cyclophosphamide; DPH, docetaxel, pertuzumab, and trastuzumab; FEC, fluorouracil, epirubicin, and cyclophosphamide; TPH, paclitaxel, pertuzumab, and trastuzumab.

**Table 4 cancers-14-02596-t004:** Summary of site of first recurrence.

Patients, *n* (%)	Cohort A (ddAC ➝ TPH) *n* = 199	Cohort B (FEC ➝ DPH) *n* = 198
**Patients with a recurrence**	15 (7.5)	25 (12.6)
**Local recurrence**	2 (1.0)	4 (2.0)
Ipsilateral after previous lumpectomy	0	2 (1.0)
Ipsilateral after previous mastectomy	1 (0.5)	2 (1.0)
**Regional recurrence**	2 (1.0)	4 (2.0)
Ipsilateral internal mammary lymph nodes	0	2 (1.0)
Ipsilateral axillary lymph nodes	2 (1.0)	2 (1.0)
Ipsilateral supraclavicular lymph nodes	0	1 (0.5)
**Distant recurrence**	13 (6.5)	13 (6.6)
Skin, subcutaneous tissue, and lymph nodes	3 (1.5)	2 (1.0)
Bone	4 (2.0)	1 (0.5)
Lung	1 (0.5)	3 (1.5)
Liver	1 (0.5)	1 (0.5)
CNS	4 (2.0)	7 (3.5)
Other	2 (1.0)	0
**Second primary invasive breast cancer**	2 (1.0)	2 (1.0)
Right	2 (1.0)	0
Left	0	2 (1.0)
**Second primary malignancy (nonbreast)**	1 (0.5)	6 (3.0)
Lung cancer	0	1 (0.5)
Colon cancer	0	2 (1.0)
Non-Hodgkin’s lymphoma	0	1 (0.5)
Other	1 (0.5)	2 (1.0)

Patients may be counted in more than one type of breast cancer recurrence category. Similarly, within breast cancer recurrence categories, patients may be counted under multiple sites/locations. CNS, central nervous system; ddAC, dose-dense doxorubicin plus cyclophosphamide; DPH, docetaxel, pertuzumab, and trastuzumab; FEC, fluorouracil, epirubicin, and cyclophosphamide; TPH, paclitaxel, pertuzumab, and trastuzumab.

## Data Availability

Qualified researchers may request access to individual patient-level data through the clinical study data request platform: https://vivli.org/. Further details of Roche’s criteria for eligible studies are available here: https://vivli.org/members/ourmembers/. For further details on Roche’s Global Policy on the Sharing of Clinical Information and how to request access to related clinical study documents, see here: https://www.roche.com/research_and_development/who_we_are_how_we_work/clinical_trials/our_commitment_to_data_sharing.htm.

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
