# Peer review of "BERENICE Final Analysis: Cardiac Safety Study of Neoadjuvant Pertuzumab, Trastuzumab, and Chemotherapy Followed by Adjuvant Pertuzumab and Trastuzumab in HER2-Positive Early Breast Cancer"

_cancers, 2022, doi:10.3390/cancers14112596_

Round 1
Reviewer 1 Report
This study reflects the results of an extensive analysis about cardiac adverse events after long-follow up on patients on treatment with double HER-2 blockage and chemotherapy (Bernice trial). Authors only described well the rates of cardiac events without any statistical analysis. In spite of the authors clearly mentioned in the methods section "no statistical hypothesis testing was planned."I miss these analysis. Moreover, some Kaplan-Meier curves were shown with any further statistical analysis (.i.e. log rank).
Other commentary is related to the methodology performed to discern the drug imputability on adverse events.
Apart from these observations, I find this article very interesting to be published.
Reviewer 2 Report
The authors present an interesting and compelling descriptive analysis of cardiac toxicity in the setting of dual-HER2 blockade. There is a large and growing body of literature raising concern regarding cardiac outcomes among those who now receive several cardio-toxic agents (anthracyclines and anti-HER2 agents). The increasing use of poly-chemotherapy regimens, in addition to radiotherapy, has given pause to some who fear adverse cardiac outcomes - the data presented here are valuable in describing the risks inherent in these regimens and allowing these risks to then be balance with overall oncologic risk for a given patients. These are valuable and compelling outcomes data.
Author Response
We thank peer reviewer 2 for the positive feedback.
Reviewer 3 Report
Dear Authors,
please see my comments in the attachment.
Best,
Reviewer
